# Wake sorting, selective predation and biogenic mixing: potential reasons for high turbulence in fish schools

Jay Willis

Turnpenny Horsfield Associates, Southampton, UK
Southampton University, International Centre for Ecohydraulics, UK

## ABSTRACT

There has been debate about animals' contribution to ocean circulation, called biomixing, or biogenic mixing. The energy input of schooling fish is significant but the eddies may be too small; so energy is dissipated as heat before impacting oceanic structure. I suggest that high turbulence caused by some very large aggregations of small animals has an important impact via a more direct ecosystem feedback process than overall ocean circulation. In the model presented here, large schools exhibit cooperative behavior and act like giant sieves grading zooplankton through individual swimmer's wakes, which focus the best prey in predictable positions. Following schoolers exploit these patterns. Then schools leave, in their wakes, chaotic turbulence enhancing growth of the smaller zooplankton and phytoplankton which has been graded out by the school. The result is a different community structure of plankton than would exist without such biomixing. Changes to plankton abundance and community structure on oceanic scales over the past century are correlated to overfishing and are consistent with this concept.

## INTRODUCTION

*Darwin (1882)* outlined a theory of ecosystem engineering on a continental scale by examination of the movement of soil by earthworms. Ecosystem engineering relates to the physical alteration of a habitat by an animal which, directly or indirectly, changes the composition of the wider environment and ecosystem, with beavers being the archetypal terrestrial example. The original suggestion for a similar biogenic mechanism in the oceans, where swimming fish and other animals move sufficient water with their bodies to significantly change the mixing of the oceans, was put forward nearly 50 years ago (*Munk, 1966*). The idea was ignored until *Huntley & Zhou (2004)* found that kinetic energy production by 11 species of schooling animals is comparable to the turbulent energy dissipation from major storms and may be 3–4 times greater than the background average rate of turbulent energy. Turbulent energy dissipation alone is not enough to characterize the impacts of turbulence on ocean structure (*Visser, 2007*). Mixing efficiency is also a measure of turbulence and it depends on the length scales of turbulent motion with

Corresponding author
Jay Willis, jkwillis@gmail.com

respect to the size of the body of liquid. Winds and tides operate at comparatively large length scales, and although their net average energy dissipation may be several times less than typical swimming animal schools, their impact on the structure of the ocean will be orders of magnitude greater (*Visser, 2007*). Several attempts have been made to resurrect the theory of biogenic mixing, based on viscous mixing (another of Darwin's ideas) (*Katija & Dabiri, 2009*) or emergent properties of schools (*Saintillan & Shelley, 2011*), as ways to suggest how swimming animals may cause significant mixing through turbulence of longer length scales. Each attempt to show biogenic mixing influencing ocean structure however looks doomed by physics (*Gregg & Horne, 2009*). Here I explore an alternative possible reason for the high levels of turbulent energy dissipation by schools of fish and suggest that it may be evidence of a more direct ecosystem feedback process.

If schooling animal's behavior is an ecosystem feedback cycle, it will share characteristics with other such cases. Herbivores maintain grassland through three basic mechanisms:

1. Selective grazing, eating the slower growing life stages or parts.
2. Targeted feedback of nutrients, including energy through mechanical disruption of soil, and distribution of seeds within defecation, and
3. Protective behavior such as eating tree saplings (*Vera, 2000*).

The presence of grass eating herbivores in an appropriate landscape results in more grass, rather than less. As a marine example: baleen whales maintained a high abundance of krill (*Willis, 2007*; *Nicol et al., 2010*) through a mix of fear and fertilization. Prey selection, targeted feedback of nutrients and protective behavior may each reinforce links of mature ecosystem feedback cycles. Fish predation structures zoo-plankton communities (*Huse & Fiksen, 2010*), and so it follows that fish and other animals that feed in large schools have evolved to optimize the abundance of their prey. Intuitive understanding of this ecological processes has, I think, led to the attractiveness of biogenic mixing as ecosystem engineering (*Visser, 2007*).

Here I suggest that intense turbulence caused by small fish which feed in large schools is part of a stronger and more more direct ecosystem feedback process than that involving general ocean circulation. Swimming fish often make a reverse von Kármán street wake pattern (Fig. 1) (*Weihs, 1973*; *Drucker & Lauder, 2002*). (A normal von Kármán street is a predictable pattern of eddies that is formed behind a stationary object in a moving fluid at certain current speeds as a stage between laminar flow and full chaotic turbulence.) The few observations which have been made of the interactions between forage fish (for instance of the family *Clupeidae*) and their prey (subclass *Copepoda*, referred to as copepods in this paper) (*Kils, 1992*) show that herring (*Clupeidae*) swim with a particular s-shaped gait when hunting which may enhance or shape the turbulent wake (*Kils, 2005*). Turbulence with eddies of similar sizes to these small fish wakes aggregates neutrally buoyant particles which are similar in size and density to adult copepods (*Ott & Mann, 2000*). Aggregation is enhanced if there are small differences in density between the particles and surrounding water, as is the case for adult copepods (*Squires & Yamazaki, 1995*; *Reigada et al., 2003*). In some cases particles can be swept into 'streaming zones'

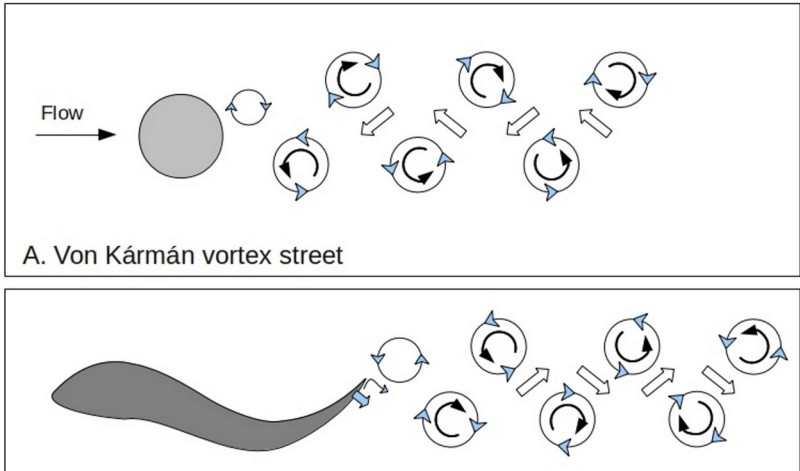

**Figure 1** **Fish wakes.** Comparison of wakes formed behind a stationary cylinder (A) and those produced by a swimming fish (B). The pattern formed by the cylinder is called a von Kármán street and consequently that of a swimming fish is called a reverse von Kármán street (*Drucker & Lauder, 2002*; *Liao, 2007*).

with up to 50 times the average density (*Wang & Maxey, 1993*). In general, larger heavier plankton are pushed to the edge of eddies (*Squires & Yamazaki, 1995*) while smaller lighter zooplankton and phytoplankton are drawn to the center (*Reigada et al., 2003*) (Fig. 2). Copepods span a critical hydrodynamic range during their life. As larvae their movements are dominated by viscous forces while as adults they are more impacted by inertial forces (Fig. 2).

## METHODS AND MATERIALS

I set up a simple hydrodynamic simulation of a reverse von Kármán street, by specifying a staggered chain of left handed and right handed eddies in a unidirectional flow field (Fig. 3). This pattern decayed through time, in the model run, as the eddies were advected across the domain and individually decayed through viscous drag. I used Fluidica (*Colonius, Taira & Merfield, 2007*) to model the flow field on a domain $524 \times 128$ square elements. Fluidica is an open-source graphical user interface (GUI) to an immersed-boundary Navier-Stokes solver (*Colonius, 2008*). The immersed-boundary method is designed to solve flows around complex-shaped and moving bodies in an incompressible fluid (*Taira & Colonius, 2007*), however, in this case I used a simple flow field to generate the pattern of a swimming fish directly. The initialized Reynolds number was 500. The model was run over 1000 time steps (approximate individual time step 0.0022 s, and domain 0.2 m $\times$ 1 m, leading to a base current $\sim$0.3 m s$^{-1}$ and eddy current $+/-$0.3 m s$^{-1}$). The Reynolds number initialization parameter in the Fluidica software relates approximately to grid based length scale, but the model was not sensitive to this parameter between a value of 100 and 1000. To this flow field I then added groups of 20,000 Lagrangian particles in randomly chosen positions within a release zone that

**Peer**J

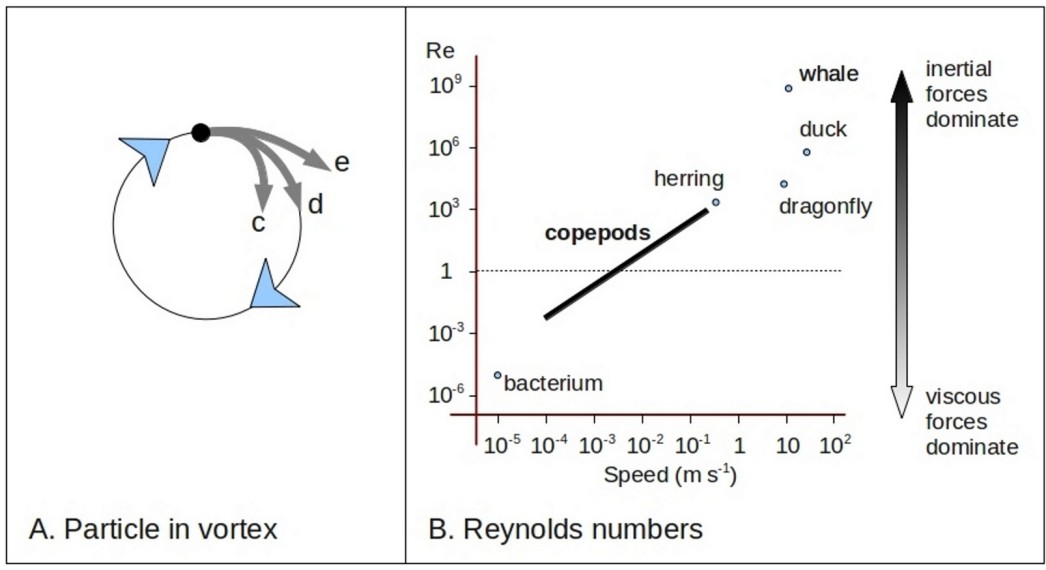

**Figure 2 Particle paths and copepod dynamics.** Trajectories of particles in a vortex depend on whether viscous forces (particle moves into vortex centre - c) or inertial forces dominate (particle spins out - e) if neither dominates the particle follows the water (d) (A). Reynolds numbers (Re) roughly characterize a situation through a ratio of inertial and viscous forces. Re is proportional to size (length scale) and speed and inversely proportional to kinematic viscosity of the fluid. Swimming copepods span a critical range dependent on their age (B), with the larva and smaller adult lives dominated by viscous forces (to left of thick black line) whereas later in life inertial forces are more important (right end of thick black line). The duck and dragonfly are in air and the others in sea water. Concept from *Tidelman (2002)*.

contained about 50% of the flow field in two counter-rotating eddies. These particles were pushed by the flow field by linearly interpolating the velocity at any point in the domain. I employed four schemes to model different sets of particles with different viscous or inertial properties. The simplest method of directly applying the interpolated velocity to each particle is called 'Forward Eulerian' (FE). This method tends to drive particles toward the edge of eddies, as it is equivalent to setting them on the trajectory of a tangent to the circular track of the water. In effect it gives the particles inertia. The standard method to overcome this artificial inertia is to use Runge-Kutta 4th order integration (RK4). RK4 involves interpolating the velocity of the particle at four points, the present position, half way to the first (FE) projected position, half way to the next projected position, and so forth. The balance of the exponents of the vector summation which follows is used to model neutrally inertial particles, the standard RK4 exponents (1/6, 1/3, 1/3, 1/6) were chosen empirically but have proven effective in field and modelling studies (*Willis, 2011*, and references therein). So I modelled neutral particles using standard RK4, and mildly inertial particles using Forward Eulerian interpolation. I modelled viscous particles by adjusting the exponents of the RK4 to drive the particles more toward the center of eddies (1/12, 1/6, 1/3, 5/12) and strongly inertial particles by using FE interpolation and carrying over 50% of velocity from previous step in FE calculation. So, for very inertial particles the velocity vector interpolated at a point was summed with the previous velocity vector with a magnitude reduced by 50% and then the particle was moved by this velocity – as such it

**Peer**J

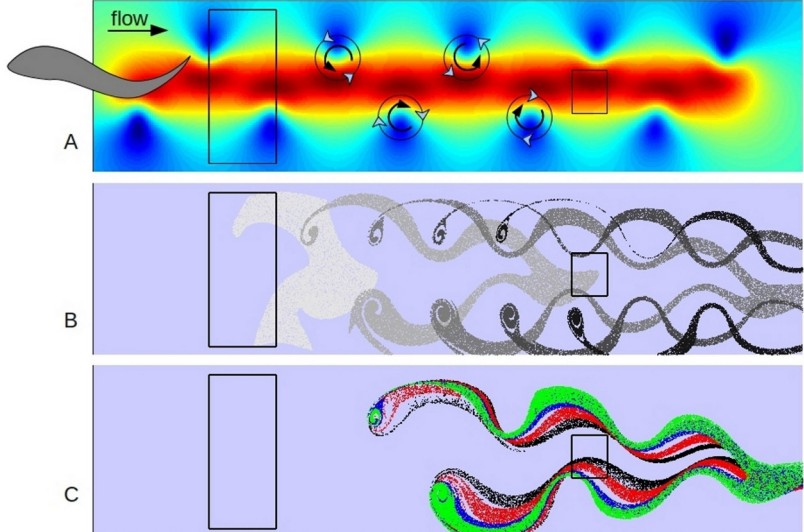

**Figure 3 Model results.** A shows the first time step of simple hydrodynamic model of reverse von Kármán street similar to that produced by a swimming fish. (The fish is only shown to illustrate approximate implied position and size.) Colors indicate speed (velocity vector magnitude), with blue close to zero and deep red close to 2 (velocity units are approximately 0.3 m s$^{-1}$), background current of one velocity unit from left to right. 20,000 particles were released in the left hand box for 100 time steps. B shows the positions of neutrally buoyant particles at 20 step intervals, starting at step 10, as they progress left to right in the model. C shows the positions of sets of 20,000 particles of different characteristics at time step 50; green are most impacted by viscous forces, blue are neutral, red are more influenced by inertial forces with black the most dependent on inertial forces. Copepods of different life stages would be similarly sorted into a spectrum based on age in a swimming fish wake. The box on the right is the target for analysis (Fig. 4) and represents a potential position of the head of a following fish in a school.

is a kind of correlated random walk (*Willis, 2011*). No provision was made for turbulence below the spatial resolution of the model, as this level of dispersion was dimensionally smaller than the other biases that had been introduced by, for instance, the approximation of inertial properties or the initialization positions.

## RESULTS

The particles were sorted highly effectively due to their different inertial properties (Fig. 4). Particles with higher inertia were swept toward the center of the eddy chain, and less inertial ones where retained in the eddies (Figs. 3B and 3C). After a time and length scale equivalent to two eddy revolutions the particles were sorted to the level of roughly 2:1. It is clear that a fish which positioned its mouth in the area directly behind another and about 1.5 lengths back would be provided with a higher concentration of higher inertial prey, and a lower concentration of lower inertial prey than if it positioned its head elsewhere.

## DISCUSSION

This study illustrates that reverse von Kármán streets are particularly well suited to selectively sort particles of different inertial properties in zones that would form immediately behind the animal producing the turbulent pattern. It uses a simple 'toy' model which is not calibrated, but nevertheless, the circumstantial evidence is strong for

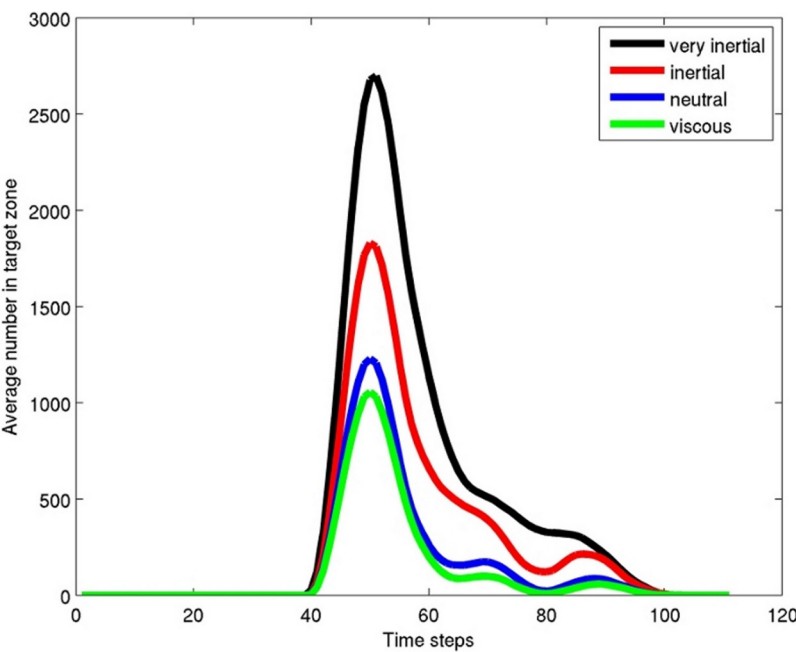

**Figure 4 Results graph.** Average (over 5 time steps) number of particles in the target box (Fig. 3) for the 100 time steps of the model. The graph shows that the particles are sorted, with many more of the inertial force dominated particles than the viscous force dominated in the target zone. Copepods would be similarly sorted in a fish wake with larger adult stages preferentially sorted into the target box and smaller life stages and other smaller items such as phytoplankton sorted out of the target zone.

this being a real effect. If fish exploit the hydrodynamics in this way, then copepod prey would be sorted in zones behind the fish according to the inertial properties of the prey. Animals following in the school will have the opportunity to position themselves to receive pulses, or streams, of various sized prey at a frequency determined by the size and tail beat frequency of the fish in front. This simple understanding leads to a fundamental shift in several basic concepts around marine ecology. For instance, it would imply that the functionality of very large schools of forage fish is based around hunting efficiency rather than only the notion of safety in numbers; fish in large predator schools might have access to a different class or quantity of prey than when alone or in small schools (this type of 'prey switch' is quite typical of terrestrial predators which hunt in groups); shape of schools and individuals' position within schools would take on different implications; and it would mean that prey (copepods) would have evolved to thrive in an environment where their vulnerability increases with inertia (which runs counter to the usual implication in marine ecosystems (*Pauly et al., 1998*)). Furthermore, the periodic introduction of turbulence impacts other participants of ocean ecosystems in important ways. Several of these implications are briefly discussed here, but they are each so fundamental that this type of model warrants further analysis. Another implication is that any devices (such as turbines, pilings or other fixed objects in strong currents) which produce a predictable turbulent eddy field may be attractive to small fish, which can take advantage of the sorting of prey at these size classes. We should expect that schooling fish wakes have been

evolved to be particularly effective in this respect, nevertheless, the environmental impact of such structures should be determined by reference to the sorting and grading potential of the wakes they produce. This effect has been identified for natural structures (some living and some physically built by animals) in currents – mainly in freshwater situations (*Vogel, 1994*) but the implications in the open ocean appears to have been largely ignored.

With respect to the key ecosystem feedback factor of prey selection: usually copepod (e.g. *Calanus* sp.) growth varies with developmental stage, strongly peaking at stages CIII and CIV, when lipid stores are being built, and tailing off at stages CIV and CV (*Harris et al., 2000*). Since all copepod adult stages target the same prey and phytoplankton (*Harris et al., 2000*), and cannibalism may be an important factor (*Eaine et al., 2002*), removing the later stages will release competitive and predatory pressure on the earlier. Indeed, fish cause the average size of plankton to reduce in marine and freshwater (*Eaine et al., 2002*; *Huse & Fiksen, 2010*) and wake sorting, described here, is likely to facilitate such selective feeding.

There are many unresolved questions: How does the escape behavior of copepods (*Waggett & Buskey, 2008*) mitigate against grading? How long before a copepod tires or gets disorientated or damaged (*Jimenez, 1997*)? And, do larger schools reinforce the turbulent effect, or dissipate it, and so where in a school is the optimal position? The shape of prey species may confound the results described here. Shape and surface roughness dictate hydrodynamic drag, and drag is increasingly important as the situation moves toward the more viscous dominated end of the Reynolds number spectrum. So copepods and other prey species may have developed physiological or behavioral adaptations to avoid being graded in a fish school through wake sorting described here. Adults may avoid being advected into the target zone by increasing their own hydrodynamic drag. When escaping a predator copepods adopt a ballistic profile; they alter their shape to reduce drag and improve inertial movement (*Waggett & Buskey, 2008*), whereas, when feeding they have feeding and swimming appendages deployed, which increase drag. There may be a balance for a relatively larger adult copepod under attack from a large school of predators between the benefits of either swimming in a ballistic mode and thus moving quickly but perhaps being sorted effectively or increasing drag by changing shape and thus moving slower but avoiding being sorted effectively. Perhaps the high mortality in boat wakes (*Bickel, Malloy Hammond & Tang, 2011*) is exacerbated by an inappropriate behavioral response to extreme turbulence.

Turbulence increases the growth rate of plankton and in general controls the plankton community structure as much the availability of nutrients, and more than environmental factors such as salinity and temperature, or external energy such as light (*Margalef, 1978*). Phytoplankton and zooplankton can be aggregated and dispersed by turbulence which can impact their future population dynamics (*Ott & Mann, 2000*; *Reigada et al., 2003*). In particular, turbulence is a critical factor in the growth and population dynamics of copepods (*Saiz & Alcaraz, 1991*; *Schmitt & Seuront, 2008*). Turbulent energy dissipation rates due to a swimming animal school can be 100-fold greater than the typical average background dissipation, and a few large schools of fish can account for up to half the total dissipation energy over a tidal cycle over a multi-km range (*Gregg & Horne, 2009*).

The characteristic length scales of this additional turbulence are in the mm to 15 cm range, much less than is responsible for oceanic mixing on the scale of winds or tides. However, the alternative is also true in that the background dissipation energy from winds and tides leaves turbulence which has much longer characteristic length scales (2–10,000 m). Thus mixing efficiency of schools of fish at scales which are relevant to plankton is orders of magnitude more important than background dissipation energy and the intensities of turbulence found in animal schools are very rare in other circumstances (*Jimenez, 1997*). Thus a large school may first act like a sieve to grade out adult copepods, as explained above, and then leave the more numerous, much smaller, but faster growing life stages with the majority of the phytoplankton, with the added benefit of otherwise rare levels of turbulence (*Saiz & Alcaraz, 1991*). If the schooling animals defecate or exude nutrients that are of value to growing phytoplankton, these will also be added to the mix.

Turbulence may provide protection for certain prey classes. Turbulence kills adult copepods at very high intensity (caused by motorboat wakes for instance) (*Bickel, Malloy Hammond & Tang, 2011*), and it is possible that similar levels of turbulent intensity are present within the wakes of schooling fish (*Gregg & Horne, 2009*; *Bickel, Malloy Hammond & Tang, 2011*) which would serve to enhance the sorting effect described above. Those animals which hunt using entrapment in long filaments, such as ctenophores or Medusa, can be damaged by even mild turbulence. Cydippid ctenophores (*Callianira antarctica*) avoid schooling krill (*Euphasia superba*) for this reason, but attack loners (*Hamner & Hamner, 2000*). Any animal which creates its own current to capture prey, such as larger adult copepods, will be disadvantaged temporarily while impacted by school turbulence. Thus the introduction of unusually intense and otherwise rare turbulence may provide protection from competition.

Finally, it is worth considering the scale over which these effects may impact actual oceans. Rough calculations based on the average packing density, swimming speed and school size suggest that the entire spawning stock of herring in the North Sea during 1963 could sweep an area of 2000 km$^2$ in 10 h to a depth of 80 m (*Huntley & Zhou, 2004*; *Gregg & Horne, 2009*; *Kenny et al., 2009*). Therefore, all North Sea herring fishing grounds could have been swept on average, from once a day, to once a week, by the standing stock of herring. Zooplankton community structure has changed over the past 100 years, during which time many fish populations have declined significantly due to overfishing, while others may have increased (*Pauly et al., 1998*; *Boyce et al., 2010*). There is some argument over this, but on coastal areas, which used to be the best fishing ground for schooling pelagic fish, all agree the change is most acute (*McQuatters-Gallop et al., 2011*). Correlations between signals of regime change suggest that the fish/plankton connectivity has been replaced by environment/plankton connectivity (*Kenny et al., 2009*) which confirms, with much other evidence (*Huse & Fiksen, 2010*), that pelagic fish impact plankton community structure on an oceanic scale.

## ACKNOWLEDGEMENTS

My thanks to the editors and referees for their helpful contributions and advice. Thanks to Mark Lorang for sharing interesting ideas.

### Funding

No funding was received for this study.

### Competing Interests

I work for Turnpenny Horsfield Associates as a consultant scientist.

### Author Contributions

- Jay Willis conceived and designed the experiments, performed the experiments, analyzed the data, contributed reagents/materials/analysis tools, wrote the paper.

### Supplemental Information

Supplemental information for this article can be found online at http://dx.doi.org/10.7717/peerj.96.

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
