# Peer review of "Wake sorting, selective predation and biogenic mixing: potential reasons for high turbulence in fish schools"

_PeerJ, doi:10.7717/peerj.96_

## Round 0.1 · original submission · Minor Revisions

Both reviewers complimented the manuscript. The paper will be a nice contribution pending your careful attention to comments by the two reviewers.

Reviewer 1 ·

Basic reporting

The manuscript tackles the challenging problem of ocean mixing produced by fish and its potential role in the "ecosystem engineering" -
sorting out the large zooplankton and facilitating by this easier grazing. The manuscript is clearly written and well structured. It presents a nice and concise overview of the state-of-the-art in the biomixing research. A simple modeling experiment presented by the author, demonstrates that the proposed hypothesis on "sorting by mixing" cannot be rejected a priori.

The study poses an interesting question and sketching the general direction, where the answer can be looked for. The manuscript presents an interdiscplinary study with focus on biology. Therefore, it might deserve eventual publication in PeerJ. However, the validity of the simple model to the real oceanic processes under simulation has to be presented in more details. I also recommend to improve the manuscript in its methodical part (see general comments). In its present form, the description of the model setup makes difficult a projection of the modeling results on the the fish-produced turbulence in the ocean.

Experimental design

The author uses a simple modeling experiment with a lot of a priori assumtions, whose reliability should be discussed more thoroughly. Some aspects of the model setup remain unclear (see general comments).

Validity of the findings

The results agree generally with the conclusions drawn in the ms. Their connection to the actual processes needs additional clarification.

Additional comments

- Line 68: The Fluidica software is not described in peer-reviewed literature. I did not find a description on the web either. Therefore, a short description of the model equations and parameterizations will help understanding its relevance to the processes modeled. I suggest, the model has no parameterization of turbulence. Is the molecular viscosity the only diffusion mechanism? How strong is the effect of numerical diffusion? How the numerical stability is ensured? In which range of Reynolds numbers the model is able to perform and how it refers to the real dissipation rates of the cohereht structures, like Karman vortex streets in the ocean? See also the next remark.

- Line 69: I could not figure out how the Reynolds number 500 resulted in currents of 1m/s. How the Reynolds is defined? If it is the domain Re, I arrive at velocities of about 0.1 mm/s. If it is the grid Re, I get velocities of ~5 cm/s (assuming the water viscosity of ~ 10^-6 m^2/s). Please, clarify.

Other remarks:

Title:
- too general, sounds like a title for a review paper and tells little about the actual content of the study. Consider reformulating to mirror the actual content of the manuscript.


Abstract:

- Lines 2-3: "has an equally important impact..." on what? And equally important to what? I can figure out from the ms that the effect of biomixing on ocean mixing seems to be negligible, but biomixing can play an important role in interaction between different levels of the ecosystem. Consider rephrasing.


- Lines 57-59 and Fig. 2: According to the text, the copepods change their flow characteristics depending on size, not speed. Why speed is chosen for the x-axis in Fig. 2? The figure seems to be completely out of the presented concept.

-Lines 70-71 and Fig. 3: The text declares current speeds to be +-1m/s. Why speeds are presented in "arbitrary units" in Fig. 3?

- Fig. 3A: presenting vorticity and/or streamlines would be more informative than absolute magnitudes of the current velocity.

·

Basic reporting

I enjoyed the article "Wake sorting, biomixing and ecosystem engineering" as it provided interesting and thought provoking ideas. There are a few distractions in the paper that could be easily changed which in my opinion would greatly improve the paper.

1) The term ecosystem engineering is over used. The fish clearly seem to be impacting ecosystem components with their swimming patterns but this impact is not intellectually planned activity of the fish. Hence, they are not engineering anything. For me the term ecosystem engineering is basically cute jargon that does not add to level of understanding. I suggest the authors simply drop the word engineering from the title and all other occurrences in the paper.

For example:
If schooling animal's behavior is an ecosystem engineering feedback cycle, it will share
characteristics with other such cases.

Could be:
If schooling animal's behavior is an ecosystem feedback cycle, it will share
characteristics with other such cases.

2) There are a couple of digs in the paper that distract from it. Some simple re-wording would be more effective.

Each attempt to show biogenic mixing influencing ocean structure however looks doomed by physics (Gregg and Horne 2009). So is biogenic mixing interesting and important, or just interesting?

Is this a direct play on words from Gregg and Horne 2009 that is totally distracting, just the author trying to be clever. Gregg and Horne 2009 provide a robust field measure of turbulence during a fish swarming event in their instrument array but they could not conclusively link those measures to understanding the dynamics within fish and zooplankton aggregation. They make conclusions supported by their data, that is simply good science.

and

Fish predation structures zoo-plankton communities (Huse and Fiksen 2010), and so it follows that fish and other animals that feed in large schools have evolved to optimize the abundance of their prey. Intuitive understanding of this ecological processes has, I think, led to the attractiveness of biogenic mixing as ecosystem engineering ('...an idea so appealing, it seems bad manners to challenge it' (Visser 2007)).

First replace the word process for the word engineering. Second Visser 2007 demonstrates in a very eloquent scaling approach that small organisms and fish can not physically cause enough mixing to global ocean circulation. Hence the comments in parenthesis need to be removed.

3) The notion that biomixing is altering ocean circulation is not even remotely addressed by the modeling presented in the paper. The modeling focuses on size selective sorting of prey for fish based on turbulent eddies produced by a swimming fish. This in of itself is very interesting but the assertions that the process may impact global ocean circulation I found to be a distraction from the main focus of the paper. I suggest to model a swarm of fish first and further test the selective sorting of prey idea rather than discuss global ocean circulation, the modeling presented simply does not scale up to be relevant to the debate.

4) If the author really wants to relate his modeling to the debate about biomixing of the ocean he will at the very least need to put his work within the context of the following list of papers and then some how scale up his results.

JO Dabiri - Geophysical Research Letters, 2010 Role of vertical migration in biogenic ocean mixing

A Lorke, WN Probst - Limnology and Oceanography, 2010 In situ measurements of turbulence in fish shoals

G Subramanian - Curr. Sci, 2010 Viscosity-enhanced bio-mixing of the oceans

K Katija - The Journal of Experimental Biology, 2012 Biogenic inputs to ocean mixing

AM Leshansky, LM Pismen - Physical Review E, 2010 Do small swimmers mix the ocean?

S Rousseau, E Kunze, R Dewey… - Journal of Physical …, 2010. On turbulence production by swimming marine organisms in the open ocean and coastal waters

Experimental design

The model set up seems to me to be appropriate and well done. A good first step worthy of publishing.

Validity of the findings

The conclusion are supported by the model results as they pertain to size selective portioning of prey by a swimming fish and how that may impact other processes like selective feeding by the fish composing a swarm, as well as impacts to size composition of the prey. It was interesting and easy to follow, thought provoking, good work.

Additional comments

I would guess that the next step in the modeling effort would be to add shape and swimming attributes to the drift particles that were "pushed" by the flow to assess outcome of those real aspects. Shape clearly impacts settling velocities of particles and is a big factor in selective entrainment, transport and deposition processes common to wind blown particles, sediment in rivers and on beaches. Organisms that swim can also sense the hydraulic forces associated with flowing water, and hence, would be expected to orient relative to the flow and or swim to reduce or enhance displacement by the turbulence created by swimming motions of the fish. A discussion of these factors would add to the paper. Could shape alone confuse the size selective displacement patterns predicted by the model?

---

## Round 0.2 · accepted · Accept

Thanks for your forthright response to reviewers. Your responses point by point did the job. This is a very nice, interesting paper.